# Systematic Review and Prevalence Meta-Analysis of Quadriceps Femoris Morphology: Significance of the Quadriceps Tendon in Anterior Cruciate Ligament Reconstruction

**DOI:** 10.3390/jfmk10030250

**Published:** 2025-06-30

**Authors:** Maria Piagkou, George Triantafyllou, Georgi P. Georgiev, George Tsakotos, Łukasz Olewnik, Ingrid C. Landfald, Bartosz Gonera

**Affiliations:** 1Department of Anatomy, School of Medicine, Faculty of Health Sciences, National and Kapodistrian University of Athens, 11527 Athens, Greece; mapian@med.uoa.gr (M.P.); georgerose406@gmail.com (G.T.); gtsakotos@gmail.com (G.T.); 2“VARIANTIS” Research Laboratory, Department of Clinical Anatomy, Masovian Academy in Płock, 09-402 Płock, Poland; lukaszolewnik@gmail.com (Ł.O.); ingridceciliee@gmail.com (I.C.L.); gonerabartosz7@gmail.com (B.G.); 3Department of Orthopedics and Traumatology, University Hospital Queen Giovanna-ISUL, Medical University of Sofia, 1527 Sofia, Bulgaria; 4Department of Clinical Anatomy, Masovian Academy in Płock, 09-402 Płock, Poland

**Keywords:** quadriceps femoris, quadriceps tendon, accessory head, accessory layer, graft harvesting

## Abstract

**Background:** The quadriceps femoris (QF) muscle is traditionally described as a four-headed structure (QF4), yet recent anatomical studies have identified significant morphological variations, including accessory heads and complex quadriceps femoris tendon (QFT) layering. These anatomical differences are especially relevant when harvesting the QFT for anterior cruciate ligament (ACL) reconstruction, where graft quality and structure are critical to surgical success. This study aimed to systematically review and quantitatively synthesize available data on QF variants, with a focus on accessory heads and tendon architecture. **Methods:** This systematic review and meta-analysis followed PRISMA 2020 and Evidence-Based Anatomy Workgroup guidelines. Cadaveric studies reporting QF variants were identified through searches of multiple databases and anatomical journals. Pooled prevalence estimates and mean QFT lengths were calculated using random-effects models. Heterogeneity and publication bias were also assessed. **Results:** Eighteen studies encompassing 1066 lower limbs met inclusion criteria. The five-headed QF (QF5) was the most common variant (54.11%), followed by the classical four-headed form (QF4) (40.74%). Rare morphologies with six to eight heads (QF6–QF8) were also documented. Among accessory heads, the vastus lateralis (VL) type was most prevalent (21.35%), while an independent tensor vastus intermedius (TVI) occurred in 13.54% of limbs. The QFT most frequently displayed a trilaminar structure (47.73%), with quadrilaminar architecture observed in 42.49%. The mean QFT length was 78.63 mm. **Conclusions:** This meta-analysis confirms that the QF often deviates from classical anatomical descriptions, frequently exhibiting supernumerary heads and multilayered tendon architecture. These findings highlight the importance of detailed preoperative imaging and personalized surgical planning to minimize complications and optimize graft selection in ACL reconstruction.

## 1. Introduction

Anterior cruciate ligament (ACL) reconstruction is one of the most common orthopedic procedures worldwide. Years of research have focused on identifying the optimal graft to reduce complications and improve postoperative outcomes. The bone-patellar tendon-bone (BPTB) and hamstring tendon (HT) grafts are the most frequently used, while the quadriceps femoris tendon (QFT) autograft has gained traction in recent years. One factor influencing the adoption of QFT may be its morphological variability, which is explored further in this review [1,2].

The musculoskeletal system exhibits both typical and variant anatomy, often studied for its clinical implications, particularly in orthopedic surgery. Anatomical variations have been documented in cadaveric and radiological studies, while systematic reviews with meta-analysis remain the gold standard for synthesizing such evidence [3,4,5].

The anterior thigh compartment typically includes the quadriceps femoris (QF) and sartorius muscles. The QF is the primary extensor of the leg and comprises four parts: rectus femoris (RF), vastus intermedius (VI), vastus lateralis (VL), and vastus medialis (VM). The RF originates from the ilium, while the vastii arise from the femur. These four parts converge into a single tendon that inserts into the base of the patella [6].

According to Gray’s Anatomy, the QF is described as a relatively consistent muscle group with minimal morphological variation [6]. In contrast, Bergman’s Comprehensive Encyclopedia of Human Anatomic Variations acknowledges the presence of occasional aberrant attachments, most commonly involving the RF [7]. These aberrations are typically limited to accessory slips or unusual origins, and notably, accessory heads of the RF have been documented only in isolated case reports, underscoring their rarity and limited clinical characterization [6,7].

Before 2016, studies on QF morphology primarily focused on the variability of the QFT due to its surgical relevance [8,9,10]. In 2016, Grob et al. identified a previously undescribed muscle head, the tensor of vastus intermedius (TVI), in all 26 lower limbs they dissected. They categorized the TVI into four morphological types: independent (entirely separate), VI-type (where the aponeurosis joins VI), VL-type (where the aponeurosis joins VL), and standard type (characterized by a common aponeurosis with VI and VL). Following this discovery, several studies have explored the presence and classification of accessory heads [11,12,13]. Olewnik et al. were the first to link these accessory heads to the layered structure of the QFT [14].

Although several narrative reviews have discussed variations in the QF [15,16,17], none have employed a systematic review and meta-analytic approach to quantify the prevalence and morphological spectrum of these variants, particularly the presence of additional heads and complex QFT layering. This study addresses that gap by synthesizing current evidence using rigorous methodology, with an emphasis on both embryological origins and clinical implications.

## 2. Materials and Methods

This systematic review and meta-analysis *were* conducted in accordance with the Evidence-Based Anatomy (EBA) Workgroup guidelines for anatomical meta-analyses [18] and the PRISMA 2020 guidelines for systematic reviews [19], following the principles established in previous studies. The protocol was not registered in any online database.

A comprehensive literature search was conducted across four major databases: PubMed, Google Scholar, Scopus, and Web of Science. The search terms were used in various combinations, including “quadriceps femoris,” “accessory head,” “additional head,” “supernumerary heads,” “layers,” “variation,” “fifth head,” “sixth head,” “supernumerary,” “tensor vastus intermedius,” “cadaveric study,” “radiological study,” and “surgical study.”

In addition to database searching, the reference lists of all included studies were screened for relevant articles. Grey literature was examined, and a targeted manual search was performed across key anatomical journals, including *Annals of Anatomy, Clinical Anatomy, Journal of Anatomy, Anatomical Record, Surgical and Radiological Anatomy, Folia Morphologica, European Journal of Anatomy, Anatomical Science International, Anatomy and Cell Biology, and Morphologie.*

Studies were included if they reported on the prevalence of QF morphological variants. *Exclusion criteria* comprised case reports, conference abstracts, animal studies, and studies lacking relevant or sufficient data.

Three independent reviewers conducted the search and extracted data into Microsoft Excel spreadsheets. Discrepancies were resolved by consensus among all authors. The risk of bias was assessed using the Anatomical Quality Assurance (AQUA) tool, developed by the EBA Workgroup for anatomical systematic reviews [19].

### Statistical Analysis

The meta-analysis was performed using the open-source R programming language and RStudio software (version 4.3.2). Analyses were conducted by a single researcher (GTr) using the “meta” and “metafor” packages.

For the meta-analysis of proportions, the pooled prevalence was estimated using the inverse variance method under a random-effects model. The Freeman–Tukey double arcsine transformation was applied to stabilize variances. Between-study variance (τ^2^) was estimated using the DerSimonian–Laird method, and confidence intervals for τ and τ^2^ were calculated using the Jackson method.

Subgroup analyses were conducted to assess the impact of potential moderators, including geographic distribution and study type, on the pooled prevalence estimates.

For continuous data (e.g., mean lengths), a separate meta-analysis was performed using untransformed (raw) means. Between-study variance was estimated using the restricted maximum-likelihood (REML) method, and the Q-profile method was applied to compute confidence intervals for τ and τ^2^.

A *p*-value of less than 0.05 was considered statistically significant. Heterogeneity across studies was assessed using Cochran’s Q statistic and quantified using the Higgins I^2^ statistic. A Q statistic with a *p*-value of less than 0.10 was considered indicative of significant heterogeneity. I^2^ values were interpreted as follows: 0–40% (not important), 30–60% (moderate heterogeneity), 50–90% (substantial heterogeneity), and 75–100% (considerable heterogeneity).

To detect potential small-study effects, a DOI plot with the Luis Furuya–Kanamori (LFK) index was generated for the meta-analysis of proportions [20].

## 3. Results

### 3.1. Studies Identification

The initial database search yielded a total of 4942 articles, which were exported to Mendeley reference manager (version 2.10.9; Elsevier, London, UK). After removing duplicates and an initial screening of titles and abstracts, 138 articles were selected for full-text review. Following full-text assessment, 16 studies met the eligibility criteria and were included in the systematic review. An additional two studies were identified through a secondary search, which included screening reference lists, reviewing grey literature, and conducting a manual search of key anatomical journals. As a result, a total of 18 studies were included in the final systematic review and meta-analysis. The study selection process is illustrated in the PRISMA 2020-compliant flow diagram (Figure 1).

### 3.2. Characteristics of the Included Studies

A total of 18 studies were included in this systematic review, comprising 1066 lower limb specimens. All studies were based on cadaveric dissection. Of these, nine studies investigated European populations, five focused on Asian populations, and four involved American populations. The detailed characteristics of the included studies are summarized in Table 1. The majority of the studies (n = 12) were published after 2016, indicating a growing interest in the QF morphological variability in recent years. Geographically, European populations were the most studied, accounting for half of the included studies, followed by Asian and American cohorts. This distribution may reflect the concentration of research activity in European Anatomy Departments. Regarding methodological quality, most studies (n = 13) were assessed as having a low risk of bias, suggesting an overall improvement in anatomical study design and reporting standards. However, earlier studies, particularly those conducted before 2010, were more likely to present a high risk of bias, often due to small sample sizes and limited methodological detail.

### 3.3. Quadriceps Femoris (QF) Heads Morphological Variability

The most prevalent QF morphology was the five-headed configuration (QF5), with a pooled prevalence of 54.11% (95% CI: 35.70–71.89). No significant differences were observed in QF5 prevalence by nationality or body side (Table 1). The DOI plot revealed minor asymmetry, with an LFK index of +1.77. The four-headed morphology (QF4) was the second most common, with a pooled prevalence of 40.74% (95% CI: 23.73–58.93). Similar to QF5, QF4 showed no significant differences across populations or body sides (Table 1), and the DOI plot demonstrated minor asymmetry with an LFK index of −1.37. Other morphologies were far less frequent: six-headed (QF6) at 3.71%, seven-headed (QF7) at 0.46%, and eight-headed (QF8) at 0.21%. Among these, only QF6 showed a significant difference across populations (*p* = 0.0140). Detailed prevalence by population and body side is presented in Table 2.

Regarding classification of the accessory head in QF5, as proposed by Grob et al. [23], the VL type was the most prevalent, with a pooled estimate of 21.35% (95% CI: 12.29–32.01). The independent type was observed in 13.54% (95% CI: 3.54–28.00), followed by the VI types at 11.90% (95% CI: 4.08–22.67). The least common was the standard type, with a pooled prevalence of 11.39% (95% CI: 1.44–27.69) (Table 3).

### 3.4. Quadriceps Femoris Tendon (QFT) Layering

The pooled mean length of the QFT was estimated at 78.63 mm (95% CI: 68.24–89.02). Among the morphological types, the trilaminar (three-layer) configuration was the most prevalent, with a pooled prevalence of 47.73% (95% CI: 6.27–90.72). However, the DOI plot showed significant asymmetry, with an LFK index of +5.03. The quadrilaminar (four-layer) morphology followed, with a pooled prevalence of 42.49% (95% CI: 5.80–84.44). Similar to the trilaminar configuration, significant asymmetry was observed in the corresponding DOI plot, with an LFK index of −3.75. Rare QFT types included the pentalaminar (five-layer) morphology, observed in 0.81% of specimens (95% CI: 0.00–7.17), and the bilaminar (two-layer) morphology, with a pooled prevalence of 0.77% (95% CI: 0.00–7.02).

## 4. Discussion

The current evidence-based systematic review and meta-analysis confirms that the five-headed quadriceps femoris (QF5) is the most common morphology, followed by the four-headed configuration (QF4). Additionally, the QFT was most frequently observed to have a trilaminar structure.

Embryologically, the lower limb begins forming at the end of the 4th developmental week (8–11 mm stage). The QF appears at approximately 11 mm as a single muscle mass in the thigh, which differentiates into four parts by the 20 mm stage, each attaching to the skeleton via distinct tendons [33,34]. Accessory heads may develop during the 11–19 mm range. After Grob et al. [23] identified the tensor of vastus intermedius (TVI), Utsunomiya et al. [35] confirmed its embryological development during Carnegie stage 22 (27 mm), noting its absence in stage 21. The TVI appears to develop from the vastus intermedius and/or vastus lateralis, supporting the classification proposed by Grob et al. [23].

Grob et al. [23] initially categorized the accessory (supernumerary) QF head, while Olewnik et al. [11] later dissected 106 lower limbs and expanded the classification:-Type I: Single independent accessory head (TVI type), 44.1%, with IA (lateral to VI) and IB (medial to VI) subtypes.-Type II: Single accessory head derived from another muscle (30.8%)-IIA (from VL), IIB (from VI), and IIC (from gluteus minimus).-Type III: Multiple accessory heads (25%)- IIIA (2 heads, common tendon), IIIB (2 heads, separate tendons), IIIC (3 heads), and IIID (4 heads).

Although QF5 is the most common, morphologies with more than five heads, such as QF6–QF8, have been sporadically reported. Grob et al. [23] documented five cases with two-headed TVIs. Olewnik et al. [23] identified six-, seven-, and eight-headed QFs under their Type III classification. Ruzik et al. [36,37], Zielinska et al. [38], and Moore et al. [39] also described unique accessory head variations.

Clinically, QFT anatomy is highly relevant in ACL reconstruction. The mean tendon length was 78.63 mm in this meta-analysis. Olewnik et al. [14] suggested harvesting up to 105 mm in males and 80 mm in females, although sex-specific data were insufficient for inclusion in this analysis. An accurate preoperative evaluation of the quadriceps femoris tendon (QFT) is essential for optimizing graft selection and harvest during ACL reconstruction. Magnetic resonance imaging (MRI) remains the gold standard due to its superior soft tissue resolution, allowing for detailed visualization of tendon length, thickness, and the number of distinct layers [8,24,32]. MRI is particularly useful in identifying variations in QF morphology, such as accessory heads [11,14] or unusual tendon architecture, which may impact graft viability and harvesting strategy. Ultrasound (US), while more operator-dependent, serves as a viable and cost-effective alternative, especially in outpatient or resource-limited settings. It provides dynamic, real-time imaging and can be effectively used to measure tendon thickness and identify gross structural anomalies [26]. Incorporating these imaging modalities into routine preoperative planning enhances anatomical accuracy, reduces intraoperative complications, and supports individualized surgical approaches that account for anatomical variability. As QF and QFT complexity become better understood, the integration of imaging into clinical workflows becomes increasingly critical for safe and effective graft harvesting.

This review found the trilaminar QFT configuration to be most common (~47%), followed closely by the quadrilaminar type (~42%). Olewnik et al. [14] noted that QFs with four layers and no accessory heads (55.5% of cases) had longer tendons (~109.55 mm), favorable for grafting. Conversely, accessory heads or layers may shorten the tendon and complicate harvesting, except in rare eight-headed QFs (~134.66 mm), deemed ideal for grafts.

The anterior midline approach is the most widely used technique for harvesting the QFT, as it provides direct access and excellent visualization of the tendon and surrounding anatomical structures [40,41]. This approach facilitates the accurate identification of the central tendon, enables the controlled harvest of full- or partial-thickness grafts, and is particularly advantageous in patients with morphological variations, such as supernumerary muscle heads or multilayered tendons [11,14,24]. In recent years, minimally invasive techniques using smaller incisions have gained popularity due to their potential to reduce soft tissue trauma, improve cosmesis, and minimize postoperative pain [42,43]. These methods are beneficial in primary ACL reconstructions with standard tendon morphology. However, the limited exposure can pose challenges in cases involving tendon bifurcation, accessory heads, or abnormal layering, which may compromise graft integrity or increase harvest time [14,32]. In contrast, extensive surgical exposure is recommended in complex or revision procedures, such as complete QFT ruptures, multi-ligament knee reconstructions, and cases with ambiguous or distorted anatomy (e.g., QF6–QF8). These situations often demand a broader field to ensure safe and complete graft retrieval, minimize the risk of inadvertent injury to surrounding structures, and verify the quality and continuity of the harvested tissue [24,26,32]. Ultimately, the choice of approach should be individualized, balancing surgical access, anatomical complexity, and patient-specific factors. Preoperative imaging with MRI or US can guide incision planning and reduce the risk of unexpected intraoperative findings [8,26,32].

ACL injuries are among the most common orthopedic injuries, particularly affecting young and physically active populations. In the United States alone, more than 100,000 ACL injuries are reported annually, representing a significant public health and economic burden [44,45]. ACL reconstruction using tendon autografts remains the gold standard for restoring knee stability and function [1,2]. Historically, BPTB and HT autografts have been the most frequently used graft types. While both provide strong fixation and good long-term outcomes, they are also associated with notable complications. BPTB grafts have been linked to anterior knee pain, patellar fractures, and an increased risk of donor-site morbidity, while HT grafts may result in hamstring weakness and altered proprioception [14,31]. In recent years, the QFT autograft has gained traction as a reliable alternative. It offers several advantages, including a large cross-sectional area, customizable thickness, and reduced donor-site morbidity [41,42,46,47,48,49]. Clinical studies have reported favorable biomechanical properties and comparable or superior functional outcomes relative to BPTB and HT grafts, making QFT an increasingly preferred choice, particularly in revision surgeries and high-demand patients.

Both partial-thickness (PT) and full-thickness (FT) QFT autografts are used in ACL reconstruction. FT grafts provide greater mechanical stability and are often preferred when maximum structural integrity is needed. However, they may increase donor-site morbidity and the risk of quadricep weakness. In contrast, PT grafts are less invasive but may have limitations in terms of tensile strength. Despite these theoretical differences, current evidence shows no significant difference in clinical outcomes between PT and FT grafts [50]. Multiple meta-analyses comparing QFT to BPTB and hamstring tendon (HT) grafts indicate comparable graft survival rates, knee stability, and functional performance [46,49,51,52,53,54]. Kanakamedala [48], in their systematic review, do not find differences in postoperative outcomes or complications regardless of the FT or PT QFT grafts. Mouarbes et al. [46] proved that the QFT autograft had better clinical and functional outcomes than the HT or BPTB autografts for ACL reconstruction. In a meta-analysis, Dai et al. [40] revealed that the QFT autograft had similar survival rates, functional results, and knee stability to the BPTB and HT autografts; these authors point out a less donor site morbidity of QT graft compared to the BPTB and HT grafts. Raj et al. [41] also depicted no significant difference between the QFT and HT autografts. Mouarbes et al. [47], in their systematic review and meta-analysis, established that the QT graft has similar outcomes and a similar survival rate compared to the BPTB and HT autografts. Moreover, this graft has less donor site pain compared to BPTB and better functional outcome compared to HT autograft. Schuster et al. [53] established a lower incidence of post-operative septic arthritis after revision ACL reconstruction with the QT graft compared to the HT graft. Nyland et al. [54], in their study, found that QT autografts had less pivot shift laxity and lower failure rates than HT autografts. Ashy et al. [55] established that the results after revision ACL reconstruction with the QT graft are comparative to HT and BTB. Meena et al. [56] also presented similar functional outcomes of QT, HT, and BPTB grafts in revision ACL reconstruction, with the hamstring graft having a higher tendency towards failure. Importantly, QFT autografts are associated with lower anterior knee pain, reduced donor-site morbidity, and higher patient satisfaction, particularly in activities involving kneeling or deep flexion. QFT autografts offer several practical and anatomical advantages:Preserve ACL synergists (e.g., hamstrings), making them ideal for athletes or patients requiring high posterior chain function.Lower kneeling pain compared to BPTB grafts, which is especially relevant for professions or lifestyles involving frequent kneeling.Customizable graft dimensions, allowing tailored width and length to patient-specific needs.Predictable intraoperative dissection, due to consistent anatomical landmarks and fewer neurovascular structures in the surgical field.QT graft has adequate histological and biomechanical properties for ACL reconstruction.QT graft has satisfactory patient-reported outcomes in primary and revision ACL reconstruction.

These factors contribute to greater surgical flexibility and patient comfort, especially in individualized or complex reconstructions.

Histological studies have shown that the QFT possesses a higher collagen content, denser fibril packing, and a larger cross-sectional area than the BPTB tendon [43,53,54,55,56,57,58,59]. These properties may enhance load-bearing capacity and reduce graft elongation or failure under repetitive stress, making QFT particularly suitable for high-demand patients. Shani et al. [60] established that the elastic modulus of the QT graft is similar to the native ACL, and the cross-sectional area of this graft is twice that of the BPTB. Moreover, the ultimate load to failure and stiffness was significantly higher for the QT graft compared to BPTB. Strauss et al. [61] reported that QT grafts with bone had similar material properties to BTB and a four-stranded semitendinosus graft. Urchek et al., after comparing a 10 mm QT graft to a six-strand HS graft, established a similar ultimate load to failure. Despite its advantages, QFT harvesting is not without risk. Potential complications include quadricep atrophy, extension weakness, and donor-site discomfort, especially when deep tendon layers are removed [31]. To address this, some authors recommend preserving the fourth (deepest) layer of the QFT, which may reduce muscle disruption while still providing an adequate graft [14].

This systematic review and meta-analysis have several limitations that should be acknowledged. First, a subset of included studies was rated as having a high risk of bias, primarily due to incomplete reporting, small sample sizes, and variable dissection techniques. These methodological inconsistencies are common in anatomical research and may affect the reliability and reproducibility of pooled estimates [19]. Second, high heterogeneity was observed in several meta-analytic outcomes, particularly in prevalence estimates of accessory heads and tendon layering patterns. This heterogeneity likely stems from differences in cadaveric populations (e.g., age, ethnicity, and preservation methods), classification systems for morphological variants, and anatomical definitions across studies. While random-effects models were used to account for this variation, the inconsistency still limits the precision of pooled findings. Third, the available data were insufficient to support sex-stratified or laterality-specific subgroup analyses. Only a minority of studies provided complete demographic details, and few stratified anatomical findings by side (left vs. right) or biological sex. This represents a significant gap, as sex and laterality may influence muscle development, tendon morphology, and graft suitability—factors that are clinically relevant for personalized surgical planning. Therefore, we encourage researchers for further studies on the QF and QFT morphology to further understand the complexity of these structures according to different nationalities, sexes, symmetry, and sidedness. Fourth, while the study adhered to PRISMA 2020 and Evidence-Based Anatomy Workgroup guidelines, publication bias cannot be entirely ruled out, especially given the inclusion of older or non-indexed anatomical literature. Studies with null or less remarkable findings may be underrepresented in the published record. Finally, the majority of included data were derived from cadaveric studies, which, although valuable, may not fully reflect in vivo anatomical variability or biomechanical behavior during dynamic movement. The extrapolation of these findings to living patients, particularly those undergoing ACL reconstruction, should be made with caution.

## 5. Conclusions

The present systematic review and meta-analysis provide strong evidence that the QF often deviates from the classic four-headed (QF4) model commonly depicted in anatomical texts. A five-headed variant (QF5), typically due to the presence of the TVI, is the most frequently observed configuration, while rare forms with up to eight heads (QF8) have also been identified. The QFT is most commonly tri- or quadrilaminar, with an average length of 78.6 mm. These structural characteristics are clinically significant, particularly in procedures such as ACL reconstruction, where QFT morphology affects graft viability and surgical outcomes. Recognizing this variability also enhances the accuracy of diagnostic imaging, especially MRI and ultrasound, by informing a more precise interpretation of soft tissue anatomy. Additionally, these findings support improvements in surgical education through better-informed cadaveric dissection, simulation-based training, and anatomical curriculum development. Further research—particularly into sex differences, side-specific variants, and functional consequences—is essential to deepen anatomical insight and guide more personalized clinical and surgical strategies.

## Figures and Tables

**Figure 1 jfmk-10-00250-f001:**
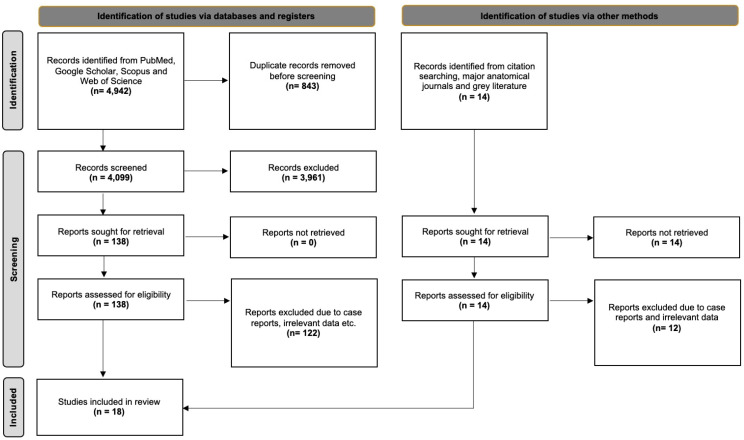
Flow chart of the search analysis, following the PRISMA 2020 guidelines.

**Table 1 jfmk-10-00250-t001:** The characteristics of the included studies.

Author (s)	Year	Population	Sample (n=)	Risk of Bias
Willan [21]	1990	European	75	High
Andrikoula et al. [9]	2006	European	10	High
Waligora et al. [10]	2009	American	20	High
Chavan et al. [22]	2016	Asian	40	Low
Grob et al. [23]	2016	European	26	Low
Grob et al. [24]	2016	European	10	High
Veeramani et al. [25]	2017	Asian	36	Low
Krebs et al. [26]	2019	American	10	High
Bonnechere et al. [27]	2020	European	20	Low
Olewnik et al. [11]	2020	European	106	Low
Sam et al. [28]	2021	Asian	41	Low
Takamura et al. [29]	2021	Asian	35	Low
Strauss et al. [30]	2021	American	18	High
Olewnik et al. [31]	2022	European	106	Low
Park et al. [12]	2022	Asian	116	Low
Olewnik et al. [14]	2023	European	128	Low
Olewnik et al. [32]	2023	European	60	Low
Oliveira et al. [13]	2024	American	81	Low

**Table 2 jfmk-10-00250-t002:** The pooled prevalences for the morphology of the quadriceps femoris (QF) variable heads among several populations and by side of occurrence.

Parameters	QF Five-Headed	QF Four-Headed	QF Six-Headed	QF Seven-Headed	QF Eight-Headed
Overall	54.11%	40.74%	3.71%	0.46%	0.21%
European	51.34%	39.82%	7.48%	1.11%	0.70%
Asian	59.91%	38.14%	1.45%	0.02%	0.00%
American	40.74%	59.26%	0.00%	0.00%	0.00%
*p*-value	0.5046	0.1560	0.0140	0.1634	0.2107
Left Side	57.19%	37.63%	0.46%	1.51%	0.46%
Right Side	66.02%	31.06%	0.54%	1.16%	0.00%
*p*-value	0.7069	0.7704	0.9528	0.9119	0.3351

**Table 3 jfmk-10-00250-t003:** The pooled prevalences and related 95% confidence intervals (CI) for the accessory heads of the five-headed quadriceps femoris (QF5) according to Grob et al. [23] classification.

Grob et al. [23] Classification Type	Pooled Prevalence (%)	95% Confidence Interval
VL type	21.35%	12.29–32.01
Independent type	13.54%	3.54–28.00
VI type	11.90%	4.08–22.67
Standard type	11.39%	1.44–27.69

## Data Availability

The data are available upon reasonable request to the corresponding author.

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
