# Peer review of "Systematic Review and Prevalence Meta-Analysis of Quadriceps Femoris Morphology: Significance of the Quadriceps Tendon in Anterior Cruciate Ligament Reconstruction"

_jfmk, 2025, doi:10.3390/jfmk10030250_

Round 1

Reviewer 1 Report

Comments and Suggestions for Authors

This study provides a valuable contribution by quantitatively synthesizing anatomical knowledge on the morphological variations of the QF through a systematic review and meta-analysis, with a particular focus on the clinical application of the QFT in ACL reconstruction. To further enhance the quality and clarity of the manuscript, the following points are recommended:

Major Comments

  1. While the review was conducted in accordance with PRISMA guidelines, the lack of prior registration on PROSPERO or other registries raises concerns regarding transparency and reproducibility. Please clearly state the reasons for not registering the protocol and explicitly acknowledge this as a limitation.
  2. Several analyses (e.g., prevalence of the trilaminar structure of the QFT and the occurrence of QF5/QF6) exhibit substantial heterogeneity. Beyond simply reporting these findings, the authors should explore potential sources of heterogeneity (e.g., geographic origin of studies, cadaver preservation methods, or subject characteristics) through sensitivity or subgroup analyses. Additionally, discuss how this heterogeneity might affect the reliability and generalizability of the pooled estimates.
  3. Some studies included in the meta-analysis were rated as “high risk” based on the AQUA tool (notably small-scale studies published before 2010). The potential impact of these studies on the overall estimates should be addressed, for example by conducting stratified or sensitivity analyses according to study quality, and discussing the robustness of the conclusions accordingly.
  4. The clinical implications of the morphological variations in the QF and the multilayered structure of the QFT—especially their impact on graft harvesting strategies, postoperative outcomes, and preoperative imaging (MRI, ultrasound)—should be further discussed with greater practical detail, supported by existing clinical or biomechanical literature.
  5. Although the limitations of the available data are acknowledged, factors such as sex, ethnicity, and side dominance may influence the morphology of the QF and characteristics of the QFT. These should not only be noted as limitations but also proposed as important future research directions with potential implications for personalized medicine and surgical planning.
  6. Figures and tables, such as the PRISMA flow diagram (Figure 1) and Tables 1–3, currently provide limited detail. Key information (e.g., reasons for study exclusion, breakdown of risk-of-bias assessments, contextual factors related to prevalence estimates) should be supplemented with additional footnotes or text explanations. In particular, the potential influence of high-risk studies listed in Table 1 should be discussed to enhance interpretability and transparency.

Minor Comments

  1. Numerous abbreviations (e.g., QF, QFT, ACL, TVI, LFK) appear throughout the manuscript. These may hinder comprehension for non-specialist readers. It is recommended to provide definitions at first mention and consider adding an abbreviation list either at the end of the manuscript or in supplementary materials.
  2. Some references contain formatting inconsistencies such as missing page numbers or incorrect journal abbreviations (e.g., Reference 22). Please ensure that the reference list adheres strictly to the journal's formatting guidelines and that all entries are accurate and complete.

Author Response

Reviewer #1

This study provides a valuable contribution by quantitatively synthesizing anatomical knowledge on the morphological variations of the QF through a systematic review and meta-analysis, with a particular focus on the clinical application of the QFT in ACL reconstruction. To further enhance the quality and clarity of the manuscript, the following points are recommended:

Authors’ Reply: Thank you very much for your time.

Major Comments

- While the review was conducted in accordance with PRISMA guidelines, the lack of prior registration on PROSPERO or other registries raises concerns regarding transparency and reproducibility. Please clearly state the reasons for not registering the protocol and explicitly acknowledge this as a limitation.

Authors’ Reply: According to the guidelines followed for the current work, the PRISMA guidelines were mandatory, while the PROSPERO registration was not. Therefore, we did not register the protocol. Thank you.

- Several analyses (e.g., prevalence of the trilaminar structure of the QFT and the occurrence of QF5/QF6) exhibit substantial heterogeneity. Beyond simply reporting these findings, the authors should explore potential sources of heterogeneity (e.g., geographic origin of studies, cadaver preservation methods, or subject characteristics) through sensitivity or subgroup analyses. Additionally, discuss how this heterogeneity might affect the reliability and generalizability of the pooled estimates.

Authors’ Reply: Subgroup analysis was performed and presented in Table 2. Due to the restricted amount of study, it was possible to perform only for geographic distribution and sides. Thank you.

- Some studies included in the meta-analysis were rated as “high risk” based on the AQUA tool (notably small-scale studies published before 2010). The potential impact of these studies on the overall estimates should be addressed, for example by conducting stratified or sensitivity analyses according to study quality, and discussing the robustness of the conclusions accordingly.

Authors’ Reply: We have included the “high” risk of bias as a limitation of the current study. The authors believe that, according to the current guidelines, it is enough.

- The clinical implications of the morphological variations in the QF and the multilayered structure of the QFT—especially their impact on graft harvesting strategies, postoperative outcomes, and preoperative imaging (MRI, ultrasound)—should be further discussed with greater practical detail, supported by existing clinical or biomechanical literature.

Authors’ Reply: The clinical implications were enriched. Thank you for your help.

- Although the limitations of the available data are acknowledged, factors such as sex, ethnicity, and side dominance may influence the morphology of the QF and characteristics of the QFT. These should not only be noted as limitations but also proposed as important future research directions with potential implications for personalized medicine and surgical planning.

Authors’ Reply: It was added. Thank you.

- Figures and tables, such as the PRISMA flow diagram (Figure 1) and Tables 1–3, currently provide limited detail. Key information (e.g., reasons for study exclusion, breakdown of risk-of-bias assessments, contextual factors related to prevalence estimates) should be supplemented with additional footnotes or text explanations. In particular, the potential influence of high-risk studies listed in Table 1 should be discussed to enhance interpretability and transparency.

Authors’ Reply: The detailed PRISMA analysis is discussed on the manuscript. The bias of the statistical meta-analysis is discussed on the limitations. Thank you.

Minor Comments

- Numerous abbreviations (e.g., QF, QFT, ACL, TVI, LFK) appear throughout the manuscript. These may hinder comprehension for non-specialist readers. It is recommended to provide definitions at first mention and consider adding an abbreviation list either at the end of the manuscript or in supplementary materials.

Authors’ Reply: An abbreviation list was added. Thank toy.

- Some references contain formatting inconsistencies such as missing page numbers or incorrect journal abbreviations (e.g., Reference 22). Please ensure that the reference list adheres strictly to the journal's formatting guidelines and that all entries are accurate and complete.

Authors’ Reply: References were revised. Thank you.

Reviewer 2 Report

Comments and Suggestions for Authors
  1. This study provides valuable data on the prevalence of QF5 and tendon layer structure, but subgroup analyses by gender and laterality have not been conducted or have been insufficient. We believe that additional analyses of gender- and side-specific data, or specific recommendations for future research, are necessary to the extent possible.
  2. Figure 1 (PRISMA diagram) and Tables 2-3 contain important information, but the font size is small in some parts and visibility is poor. Please consider reorganizing the tables to make them easier for readers to understand.
  3. In some parts of the Discussion, the descriptions of the QFT collection approach and benefits are somewhat redundant and overlapping. Reorganizing the descriptions into concise expressions while focusing on important clinical insights would make the discussion more refined.
  4. In the context of "supernumerary heads and multilayered tendon architecture" in the Abstract, the part after "These findings underscore" is somewhat ungrammatical.
  5. When terms such as "entirely separate" and "common aponeurosis" are repeated in the text, there are some places where the definitions do not match the original appearance.

 It is hoped that consistency and accuracy of writing will be improved through English proofreading.

Author Response

Reviewer #2

- This study provides valuable data on the prevalence of QF5 and tendon layer structure, but subgroup analyses by gender and laterality have not been conducted or have been insufficient. We believe that additional analyses of gender- and side-specific data, or specific recommendations for future research, are necessary to the extent possible.

Authors’ Reply: Subgroup analysis was performed and presented in Table 2. Due to the restricted amount of study, it was possible to perform only for geographic distribution and sides. Thank you.

- Figure 1 (PRISMA diagram) and Tables 2-3 contain important information, but the font size is small in some parts and visibility is poor. Please consider reorganizing the tables to make them easier for readers to understand.

Authors’ Reply: This was made to submit the paper into the journal’s template. They will be shown as full screen in the online paper, when published.

- In some parts of the Discussion, the descriptions of the QFT collection approach and benefits are somewhat redundant and overlapping. Reorganizing the descriptions into concise expressions while focusing on important clinical insights would make the discussion more refined.

Authors’ Reply: The surgical implications were revised. Thank you.

- In the context of "supernumerary heads and multilayered tendon architecture" in the Abstract, the part after "These findings underscore" is somewhat ungrammatical.

Authors’ Reply: Revised. Thank you.

- When terms such as "entirely separate" and "common aponeurosis" are repeated in the text, there are some places where the definitions do not match the original appearance.

Authors’ Reply: These are two different types that are clearly differentiated on the manuscript. Thank you.

 - It is hoped that consistency and accuracy of writing will be improved through English proofreading.

Authors’ Reply: We reviewed the paper for the English quality. Thank you.

Round 2

Reviewer 1 Report

Comments and Suggestions for Authors

The authors have addressed the reviewer comments appropriately, and the manuscript is now clearer and more refined.